# Pesticides Contamination of Cereals and Legumes: Monitoring of Samples Marketed in Italy as a Contribution to Risk Assessment

Valeria Nardelli [1], Valeria D'Amico [1], Mariateresa Ingegno [1], Ines Della Rovere [1], Marco Iammarino [1,*], Francesco Casamassima [1], Anna Calitri [1], Donatella Nardiello [2], Donghao Li [3] and Maurizio Quinto [2,3,*]

1 Istituto Zooprofilattico Sperimentale della Puglia e della Basilicata, Via Manfredonia 20, 71121 Foggia, Italy; valeria.nardelli@izspb.it (V.N.); valeria.damico@izspb.it (V.D.); mariateresa.ingegno@izspb.it (M.I.); ines.dellarovere@izspb.it (I.D.R.); francesco.casamassima@izspb.it (F.C.); anna.calitri@izspb.it (A.C.)
2 Dipartimento di Scienze Agrarie, Alimenti, Risorse Naturali e Ingegneria—Università degli Studi di Foggia, Via Napoli, 25, 71122 Foggia, Italy; donatella.nardiello@unifg.it
3 Department of Chemistry, Yanbian University, Park Road 977, Yanji 133002, China; dhli@ybu.edu.cn
* Correspondence: marco.iammarino@izspb.it (M.I.); maurizio.quinto@unifg.it (M.Q.)

**Featured Application: This work offers a contribution to risk assessment regarding the levels of 37 pesticides in cereal and legume samples commercialized in Italy during the last years. It is well-known that prolonged exposure to pesticides can increase the risk of cardiovascular and respiratory disease, other than promoting cancer diseases. Thus, the World Health Organization and the European Food Safety Authority ask for monitoring of the levels of such substances, especially in vegetables, continuously, to have availability updated and detailed data on this type of food contamination. This study represents a valid contribution to risk assessment, since it is based on fully validated and accredited analytical procedures, and it supplies accurate data related to more than 200 samples of widely-consumed cereal and legume types.**

**Abstract:** The evaluation of cereal-based product contamination by pesticide residues is a topic of worldwide importance, and reliable analytical methods for official check analyses and monitoring studies are required for multi-residue analysis at trace levels. In this work, a validated multi-residual analytical method by gas-chromatography and tandem mass spectrometry coupled with a rapid QuEChERS procedure was used for the determination of 37 pesticides (pyrethroids, organophosphorus and organochlorine compounds) in 209 commercially available samples of cereals and 11 legumes, placed on the Italian market in 2018 and 2019, coming from different regions of Italy, eastern Europe, and some non-European countries. No pesticide traces were observed in the analyzed legume samples. A total of 18 cereal samples were found to be contaminated by at least one pesticide, with a concentration level higher than the corresponding quantification limit, but never exceeding the maximum level fixed in the European Regulations. This work is the first part of a surveillance study for pesticide control in food samples.

**Keywords:** pesticides; cereals; legumes; food safety; food contamination; gas-chromatography and mass spectrometry (GC-MS)

## 1. Introduction

According to the World Health Organization (WHO), pesticides are "chemical compounds that are used to kill pests, including insects, rodents, fungi and unwanted plants (weeds)". Over 1000 different pesticides are used around the world. Pesticides are used in public health to kill vectors of disease, such as mosquitoes, and in agriculture to kill pests that damage crops. Pesticides can be classified into different groups by target organism (e.g., herbicides, insecticides, fungicides, rodenticides, and pediculicides), chemical

structure (e.g., organic, inorganic, synthetic, or biological), and physical state (e.g., gaseous, fumigants). Among the others, organochlorines and organophosphates represent two of the most prominent pesticide families. Organochlorine pesticides (OCPs) are chlorinated hydrocarbons, extensively employed in agriculture and insect control from the 1940s to the 1960s. OCPs have been banned in most of the technologically advanced countries and replaced by other synthetic insecticides, such as organophosphate pesticides (OPPs) in the 1960s and pyrethroids (PYRs) in the 1980s, due to their significant toxicity to plants and animals, including humans, and to their persistence and potential to bioaccumulation. OPPs are widely used: 50% of the killing agents used in agriculture belong to this compound class [1]. Several studies reported that prolonged exposure to OPPs can increase the risk of cardiovascular and respiratory disease, and promote cancer diseases [2]. For these reasons, OPPs have been banned in most residential uses, but they are still allowed as pesticides on fruits and vegetables. Pyrethroids are synthetic analogs of pyrethrins, natural insecticides produced by certain species of chrysanthemum. PYRs are commonly used in crop protection, veterinary medicine, and for indoor/outdoor pest control [3]. PYRs are generally less toxic toward humans than OCPs and OPPs, but their stability and persistence in the environment make possible their transfer along the food chain, becoming consequently a health and environmental concern.

In the last decades, in the European Community, pesticide legislation in foodstuffs and animal feeds has been continuously revised, leading to a continued update of maximum residue levels (MRLs) admitted for pesticides, based on their toxicity and the consumption of potentially contaminated food products. Specific official documents (classified by pesticide class and type of food sample) and related updates are online available [4]. Risks related to pesticide contaminations and the consumption of large amounts of cereals and legumes worldwide have promoted the development of a variety of analytical methods for their determination. Due to their high volatility, the analytical determination of pesticides is usually accomplished by using gas-chromatography. Electron capture detector (ECD), nitrogen-phosphorus detector (NPD) and flame ionization detector (FID) have been usually employed for pesticide determination, but each of them is specific for determinate pesticide class, making the pesticide comprehensive determination unfeasible [5–7]. Consequently, their use has been largely replaced by mass spectrometry, due to its higher selectivity, sensitivity and applicability to high number of pesticides [8,9].

In this work, a multiresidual method has been validated for the analysis of pyrethroids (phenothrin, permethrin, cyfluthrin, cypermethrin, deltamethrin, fenvalerate), organochlorine pesticides (aldrin, dieldrin, endrin, $\alpha$-HCH, $\beta$-HCH, $\gamma$-HCH, $\alpha$-endosulfan, $\beta$-endosulfan, endosulfan sulphate, cis-chlordane, trans- chlordane, heptachlor, heptachlor exo-epoxide, heptachlor endo-epoxide, p,p' DDD, p,p' DDE, p,p' DDT, quintozene, tecnazene) and organophosphorus compounds (azinphos-ethyl, chlorfenvinphos, chlorpyrifos, chlorpyrifos-methyl, diazinon, fenthion, malathion, parathion-ethyl, pirimiphos-methyl, profenofos, pyrazophos, triazophos) through the evaluation of linearity, selectivity, detection, and quantification limits, recovery, precision, and measurement uncertainty. Then, the evaluation of the contamination grade by OCPs, OPPs, and PYRs in more than 200 samples of cereals and legumes, coming from Italy and foreign countries and collected in the years 2018–2019 has been carried out. Considering the high number of samples and their widespread origin, the obtained data can be considered a first preliminary surveillance study that can contribute to the knowledge of food pesticide contamination.

## 2. Materials and Methods

### 2.1. Chemicals

Certified Reference Material (CRM) at a concentration of 100 mg $L^{-1}$ in acetonitrile or toluene (see Table 1 for details) of pyrethroids (phenothrin, permethrin, cyfluthrin, cypermethrin, deltamethrin, fenvalerate), organochlorine pesticides (aldrin, dieldrin, endrin, $\alpha$-HCH, $\beta$-HCH, $\gamma$-HCH, $\alpha$-endosulfan, $\beta$-endosulfan, endosulfan sulphate, cis-chlordane, trans-chlordane, heptachlor, heptachlor exo-epoxide, heptachlor endo-epoxide,

p,p′ DDD, p,p′ DDE, p,p′ DDT, quintozene, tecnazene) and organophosphorus compounds (azinphos-ethyl, chlorfenvinphos, chlorpyrifos, chlorpyrifos-methyl, diazinon, fenthion, malathion, parathion-ethyl, pirimiphos-methyl, profenofos, pyrazophos, triazophos) were purchased by a certified supplier, as requested by ISO 34 and ISO 17025 guides (Lab Instruments, Castellana Grotte, BA, Italy). Intermediate standards solutions were prepared in isooctane just before injection by diluting stock solution to obtain concentrations of 0.025–0.050–0.100–0.250 and 0.500 mg L$^{-1}$ for each pesticide. Standard working solutions were stored at $-20$ °C and used for not more than a week (as suggested by stability tests performed on the calibration standards in solution). Polychlorinated biphenyl (PCB) 209 (purity > 99.0%; 10 mg L$^{-1}$ in isooctane, Dr Ehrenstorfer, Augsburg, Germany) was used as an internal standard and added to pesticide standard calibration solutions to a final concentration of 0.100 mg L$^{-1}$. QuEChERS QuE-Lab® EN15662 Citrate LLE Tube and QuE-Lab® EN15662 PSA/C18 dSPE Tube were supplied by Lab Instruments (Castellana Grotte, BA, Italy). Acetonitrile was purchased from Sigma-Aldrich Co. (St. Louis, MO, USA), while isooctane from Panreac Química S.L.U. (Castellar del Vallès, Barcelona, Spain).

**Table 1.** List of pesticides analyzed by GC-MS/MS.

| Pesticide | Class | CAS Number | Molecular Weight | Solubilization Solvent | Precursor Ions | Diagnostic Ions |
|---|---|---|---|---|---|---|
| Aldrin | organochlorine | 309-00-2 | 364.9 | Toluene | 262.9 | 190.9; 192.9 |
| Azinphos-ethyl | organophosphorus | 2642-71-9 | 345.4 | Acetonitrile | 132.0 | 51.0; 77.0 |
| cis-Chlordane | organochlorine | 5103-71-9 | 409.8 | Toluene | 271.7; 372.8 | 236.8; 265.9 |
| trans–Chlordane | organochlorine | 5103-74-2 | 409.8 | Toluene | 271.7; 372.8 | 236.8; 265.9 |
| Chlorfenvinphos | organophosphorus | 470-90-6 | 359.6 | Acetonitrile | 266.9 | 159.0; 203.0 |
| Chlorpyrifos | organophosphorus | 2921-88-2 | 350.6 | Acetonitrile | 314.0 | 258.0 |
| Chlorpyrifos-methyl | organophosphorus | 5598-13-0 | 322.5 | Acetonitrile | 125.0; 286.0 | 79.0; 93.0 |
| Cyfluthrin | pyrethroid | 1820573-27-0 | 434.3 | Toluene | 206.0 | 151.0; 177.0; 179.0 |
| Cypermethrin | pyrethroid | 52315-07-8 | 416.3 | Toluene | 181.0 | 127.0; 152.0 |
| p,p′ DDD | organochlorine | 72-54-8 | 320.0 | Toluene | 235.0; 237.0 | 165.0 |
| p,p′ DDE | organochlorine | 72-55-9 | 318.0 | Toluene | 246.0; 318.0 | 176.0; 248.0 |
| p,p′ DDT | organochlorine | 50-29-3 | 354.5 | Toluene | 235.0; 237.0 | 165.0 |
| Deltamethrin | pyrethroid | 52918-63-5 | 505.2 | Acetonitrile | 181.1; 252.8 | 152.0; 92.9 |
| Diazinon | organophosphorus | 333-41-5 | 304.3 | Acetonitrile | 137.1 | 54.1; 84.1 |
| Dieldrin | organochlorine | 60-57-1 | 380.9 | Toluene | 262.8; 277.0 | 227.8; 241.0 |
| α-Endosulfan | organochlorine | 959-98-8 | 406.9 | Toluene | 195.0; 240.0 | 160.0; 206.0 |
| β-Endosulfan | organochlorine | 33213-65-9 | 406.9 | Toluene | 195.0; 240.9 | 160.0; 206.0 |
| Endosulfan sulphate | organochlorine | 1031-07-8 | 422.9 | Toluene | 238.7; 271.8 | 203.9; 234.9 |
| Endrin | organochlorine | 72-20-8 | 380.9 | Toluene | 245.0; 262.9 | 173.0; 193.0 |
| Fenthion | organophosphorus | 55-38-9 | 278.3 | Acetonitrile | 245.3; 278.0 | 125.0; 109.0; 169.0 |
| Fenvalerate | pyrethroid | 51630-58-1 | 419.9 | Acetonitrile | 125.0; 167.0 | 83.3; 125.0 |
| α-HCH | organochlorine | 319-84-6 | 290.8 | Toluene | 181.0; 219.0 | 145.0; 183.0 |
| β-HCH | organochlorine | 319-85-7 | 290.8 | Toluene | 181.0; 219.0 | 145.0; 183.0 |
| γ-HCH | organochlorine | 58-89-9 | 290.8 | Toluene | 180.9; 218.9 | 144.0; 182.9 |
| Heptachlor | organochlorine | 76-44-8 | 373.3 | Toluene | 99.8; 272.0 | 65.0; 237.0 |
| Heptachlor exo-epoxide | organochlorine | 1024-57-3 | 389.3 | Toluene | 262.9; 352.8 | 192.9; 262.9 |
| Heptachlor endo-epoxide | organochlorine | 28044-83-9 | 389.3 | Toluene | 183.0 | 119.0; 155.0 |
| Malathion | organophosphorus | 121-75-5 | 330.4 | Acetonitrile | 158.0; 173.1 | 125.0; 99.0 |
| Parathion-ethyl | organophosphorus | 56-38-2 | 291.3 | Acetonitrile | 109.0; 291.0 | 81.0; 109.0 |
| Permethrin | pyrethroid | 52645-53-1 | 391.3 | Toluene | 183.0 | 128.0; 152.0; 168.0 |
| Phenothrin | pyrethroid | 26002-80-2 | 350.4 | Acetonitrile | 183.0 | 115.0; 128.0 |
| Profenofos | organophosphorus | 41198-08-7 | 373.6 | Acetonitrile | 296.7; 336.9 | 268.9; 266.9 |
| Pyrazophos | organophosphorus | 13457-18-6 | 373.4 | Acetonitrile | 221.0 | 148.7; 193.1 |
| Pirimiphos-methyl | organophosphorus | 29232-93-7 | 305.3 | Acetonitrile | 290.1 | 125.0; 233.0 |
| Quintozene | organochlorine | 82-68-8 | 295.3 | Toluene | 213.8 | 141.9; 178.9 |
| Tecnazene | organochlorine | 117-18-0 | 260.9 | Toluene | 214.8 | 143.6; 178.7 |
| Triazophos | organophosphorus | 24017-47-8 | 313.3 | Acetonitrile | 161.0 | 105.7; 134.1 |

## 2.2. Sampling and Sample Preparation

A total of 209 samples of cereals (wheat: 179; pasta: 5; bran: 1; barley: 13; oats: 6; spelt: 2; corn: 2; rice: 1) and 11 legumes (beans: 8; peas: 2; chickpeas: 1) were analyzed in the Chemistry Department of the Istituto Zooprofilattico Sperimentale della Puglia e della Basilicata (Foggia, Italy), during the years 2018–2019. Several wheat samples were picked up by Italian law enforcement during the official control operations of ships coming from abroad. Other samples of cereals, pasta, and legumes were collected from local farms, grain storage warehouses, mills, and markets, regularly inspected by health services. A representative portion of each sample of cereals or legumes was thoroughly ground using a food processor. A 5 g portion was placed in a 50-mL tube and blended with 10 mL of distilled water by vortex for 1 min, and then 10 mL of acetonitrile were added. For the QuEChERS extraction, the citrate buffer was used, composed of magnesium sulfate anhydrous (4 g), sodium chloride (1 g), trisodium citrate dihydrate (1 g), and sodium citrate dibasic sesquihydrate (0.5 g). After agitation by vortex for 1 min and centrifugation for 5 min at 3000 rpm, an aliquot of 6 mL of organic phase extract was transferred in a Falcon$^{TM}$ tube containing 0.150 g of C18 EC (end-capped), 0.150 g of primary secondary amine (PSA) and 0.900 g of magnesium sulfate anhydrous (QuEChERS PSA/C18 dSPE). After agitation for 1 min and centrifugation for 5 min at 3000 rpm, 5 mL of the clear supernatant were collected and evaporated under a stream of nitrogen at 45 °C by a Turbovap system (Caliper Mod. LV, Hopkinton, MA, USA). Finally, the residue was dissolved in 1 mL of a PCB 209 solution at a concentration of 0.100 mg $L^{-1}$ in isooctane and then injected into the GC-MS/MS system. Analyses were performed in duplicate.

## 2.3. Gas Chromatography/Mass Spectrometry Analyses

GC-MS/MS analyses were performed on a Thermo Scientific TSQ EVO 8000 GC system equipped with a triple quadrupole mass spectrometer (Thermo Fisher Scientific, Waltham, MA, USA). The temperature of the ion source and transfer-line were 260 °C and 250 °C, respectively. Gas chromatographic analysis was carried out in the monitoring reaction mode. The presence of at least two significant MS/MS transitions was used to identify analytes. For each pesticide, the m/z values for the MS/MS transitions have been fixed based on what was reported in the official European documents (SANTE 2017/11813/EC and Dec 2002/657/EC). The ion selection was performed by choosing characteristic isotopic ions, especially Cl clusters, not exclusively originating from the same part of the analyte molecule. The selected diagnostic ions are shown in Table 1. The chromatographic separations were performed using the capillary column Rxi (30 m × 0.25 mm × 0.25 μm) from RESTEK Pure Chromatography (Bellefonte, PA, USA). A sample volume of 1.5 μL was injected by programmed temperature vaporizing (PTV) in splitless mode. The injector temperature started at 70 °C and after 0.05 min ramped to 260 °C at a rate of 5 °C s$^{-1}$. After 1 min, a cleaning step of 5 min at 320 °C was applied. The oven temperature was initially set at 70 °C for 1.0 min and then increased to 150 °C at a rate 30 °C min$^{-1}$ and to 260 °C at 6 °C min$^{-1}$; a final temperature of 290 °C, reached up at a rate of 20 °C min$^{-1}$, was kept for 5.0 min with a total run time of 28.0 min. The flow rate of the carrier gas (Helium, 99.999%, pressure-pulse mode: 30 psi for 1 min) was 1.0 mL min$^{-1}$. Acquisition and data processing were performed by the Trace Finder and Xcalibur workstations (Thermo Fisher Scientific).

## 2.4. Risk Exposure

The pesticide content in cereal and legume samples was evaluated by interpolation on the corresponding external standard calibration curves. Most of the analyzed samples showed no quantifiable residues of pesticides, therefore the risk exposure was studied considering only the pesticides detected with content above the corresponding quantification limit found in wheat (durum and soft) and oats samples, i.e., chlorpyrifos, fenvalerate, cyfluthrin, phenothrin, deltamethrin, cypermethrin, and pirimiphos-methyl. Regarding the food type and the related human mean consumption, the following cereal-based products were considered: pasta for durum wheat sample, bread for soft wheat, and breakfast cereals

for oats samples. Then, the data related to the pesticide occurrence in cereals were corrected on the basis of the mean percentage of the raw material in the product. The following corrective percentages were then applied: 65% for soft wheat in bread, 85% for durum wheat in pasta, and 90% for oats in breakfast cereals [10,11].

Considering the specific normative [12–19], the risk exposure was evaluated taking into account the relevant 2-years no observed adverse effect level (NOAEL) and the admissible daily intake (ADI). The toxic effect of different active substances, and the corresponding NOAELs and ADIs, were studied on rats/mice: therefore, both parameters were elaborated following the indications reported in the official European Food Safety Authority document [20]. In this report, the default values to be used in the absence of actual measured data are proposed. The conversion factors applied in the present study were as follows: body weight for children and adolescents = 12 kg; body weight for adults and elderly = 70 kg; 0.05 is the default factor to convert a feed concentration of test substances (mg kg$^{-1}$) into a daily dose for rats (mg kg$^{-1}$ b.w.) for chronic studies.

Regardless of their country of origin, all the samples monitored during this survey were collected in Italy. The reference data related to Italian food consumption were found in the INRAN-SCAI 2005-06 report [21], where the mean consumption of the three categories of cereal-based products (pasta, bread, and breakfast cereals) by this population is provided. In this document, data are divided into five population subgroups: infants (0–2 years), children (3–9 years), adolescents (10–17 years), adults (18–64 years), elderly (65–97 years). Anyway, in the present risk assessment study, only the last 4 subgroups were taken into account, since data available for the infants were considered not representative (n < 30) [22,23].

The percentage of NOAEL and ADI were calculated according to the following Equation (1):

$$\% \text{ NOAEL (or \% ADI)} = \frac{x \cdot p \cdot c}{1000 \cdot NOAEL \, (or \, ADI)} \cdot 100 \tag{1}$$

where:

$x$ is the pesticide concentration quantified in the sample
$p$ is the conversion factor related to the cereal percentage in the product
$c$ is the mean food consumption according to INRAN-SCAI 2005-06 report

## 3. Results

### 3.1. Method Validation

As recommended by the European regulations (European Commission SANTE 2017/11813/EC, European Commission Regulation 2017/644/EC and European Commission Decision 657/2002/EC), the method validation is an essential prerequisite to provide accurate and reliable results during the official monitoring and risk-assessment studies. Moreover, laboratories in charge of food and animal feed control need analytical methods able to identify/quantify the highest number of compounds within the same analysis, to optimize both times and costs. In this regard, several analytical approaches were developed during the last years for the multi-detection of environmental contaminants [24], drug residues [25], heavy metals [26], and other toxic compounds that may be present in food. Therefore, this GC-MS/MS analytical method was coupled to a rapid QuEChERS procedure for the multi-residue pesticide analysis. The method was validated through the evaluation of linearity, detection and quantification limits, selectivity, precision, recovery, and measurement uncertainty.

The linearity test was performed by the evaluation of correlation coefficients of the calibration curves obtained by the ratio between the analyte peak area and IS peak area vs. the pesticide concentration in the range 0.025–0.500 mg L$^{-1}$. For all pesticides, the calculated correlation coefficients were always higher than 0.9900. The signal-to-concentration ratio (y/x) was calculated for each experimental point to evaluate the goodness-of-fit of the data to the calibration curve. Then, the $x_i/y_i$ ratios were checked to ensure that their

deviation from the mean value of the signal-to-concentration ratio never exceeded $\pm 10\%$. The absence of systematic instrumental bias was confirmed by the confidence interval for the intercept that included the zero value at 95% confidence level. By Mandel's fitting test [27] the residual variances, resulting from the linear and the quadratic calibration function, were compared by an F-test and the hypothesis $H_0$ (no significant difference between the residual variances) was accepted for all pesticides. Therefore, calibration straight-lines rather than over curvilinear or non-linear models well fitted the experimental data.

The instrumental detection (LODs) and quantification (LOQs) limits were calculated by standard solution analysis, according to the following equations: $LOD = 3.3 s_a/b$; $LOQ = 10 s_a/b$, where $s_a$ was the standard deviation of the intercept and b the slope of the regression line obtained from the calibration curve [28]. LODs and LOQs were in the range 0.015–0.198 µg L$^{-1}$ and 0.045–0.599 µg L$^{-1}$, respectively. Similarly, LODs and LOQs in the matrix were evaluated by matrix-matched standard calibration in wheat blank samples at concentrations of 0.025–0.050–0.100–0.250 and 0.500 mg kg$^{-1}$ for each pesticide. In the matrix, LODs and LOQs ranging from 0.018–0.168 µg kg$^{-1}$ and 0.053–0.510 µg kg$^{-1}$ were obtained, as reported in Table 2. These values, considerably lower than the legal limits established by the European Community for the pesticide residues in cereal samples, confirmed the high sensitivity of the described method at trace levels, reducing the risk of false-negative results.

**Table 2.** Performance and chromatographic parameters of pesticides analyzed by GC-MS/MS.

| Analyte | $t_R$ [a] (min) | LOD µg L$^{-1}$ | LOQ µg L$^{-1}$ | LOD µg kg$^{-1}$ | LOQ µg kg$^{-1}$ | Recovery $\pm$ SD [d] | Uncertainty |
|---|---|---|---|---|---|---|---|
| | | Solvent [b] | | Matrix: Wheat [c] | | (%) | (%) |
| Aldrin | 14.47 | 0.138 | 0.418 | 0.154 | 0.466 | 74.8 ± 8.6 | 11.5 |
| Azinphos-ethyl | 23.26 | 0.080 | 0.243 | 0.086 | 0.259 | 110.3 ± 6.5 | 10.2 |
| cis-Chlordane | 16.85 | 0.109 | 0.329 | 0.083 | 0.251 | 70.6 ± 7.7 | 23.1 |
| trans-Chlordane | 16.40 | 0.115 | 0.349 | 0.143 | 0.381 | 77 ± 12 | 23.8 |
| Chlorfenvinphos | 15.90 | 0.020 | 0.060 | 0.147 | 0.446 | 96.3 ± 6.1 | 15.3 |
| Chlorpyrifos | 14.66 | 0.015 | 0.045 | 0.057 | 0.173 | 97.7 ± 8.7 | 16.2 |
| Chlorpyrifos-methyl | 13.22 | 0.129 | 0.389 | 0.093 | 0.282 | 91.8 ± 6.0 | 15.5 |
| Cyfluthrin | 24.57 | 0.063 | 0.192 | 0.122 | 0.369 | 107 ± 15 | 3.3 |
| Cypermethrin | 25.05 | 0.081 | 0.245 | 0.168 | 0.510 | 109 ± 11 | 14.6 |
| p,p′ DDD | 18.77 | 0.048 | 0.144 | 0.114 | 0.344 | 78.3 ± 4.8 | 7.4 |
| p,p′ DDE | 17.49 | 0.130 | 0.394 | 0.025 | 0.077 | 74.9 ± 5.0 | 13.9 |
| p,p′ DDT | 19.87 | 0.186 | 0.565 | 0.101 | 0.306 | 71 ± 10 | 5.9 |
| Deltamethrin | 27.25 | 0.152 | 0.460 | 0.018 | 0.056 | 104 ± 14 | 6.7 |
| Diazinon | 11.78 | 0.107 | 0.324 | 0.037 | 0.113 | 97 ± 11 | 20.2 |
| Dieldrin | 17.54 | 0.148 | 0.450 | 0.035 | 0.106 | 77.2 ± 6.6 | 13.7 |
| α-Endosulfan | 16.76 | 0.111 | 0.337 | 0.106 | 0.322 | 79.6 ± 5.3 | 15.5 |
| β-Endosulfan | 18.50 | 0.134 | 0.406 | 0.078 | 0.237 | 83.0 ± 5.3 | 9.9 |
| Endosulfan sulphate | 19.77 | 0.174 | 0.527 | 0.121 | 0.367 | 92.6 ± 8.1 | 16.8 |
| Endrin | 18.20 | 0.170 | 0.515 | 0.160 | 0.484 | 83 ± 14 | 15.4 |
| Fenthion | 14.62 | 0.035 | 0.107 | 0.068 | 0.207 | 86 ± 11 | 10.1 |
| Fenvalerate | 26.34 | 0.038 | 0.115 | 0.117 | 0.353 | 104 ± 11 | 13.2 |
| α-HCH | 10.48 | 0.151 | 0.458 | 0.075 | 0.226 | 81.3 ± 9.7 | 7.8 |
| β-HCH | 11.36 | 0.111 | 0.335 | 0.061 | 0.186 | 81.6 ± 9.1 | 10.2 |
| γ-HCH | 11.40 | 0.142 | 0.430 | 0.084 | 0.255 | 82 ± 10 | 8.6 |

**Table 2.** *Cont.*

| Analyte | $t_R$ [a] (min) | LOD µg L$^{-1}$ | LOQ µg L$^{-1}$ | LOD µg kg$^{-1}$ | LOQ µg kg$^{-1}$ | Recovery ± SD [d] | Uncertainty |
|---|---|---|---|---|---|---|---|
| | | Solvent [b] | | Matrix: Wheat [c] | | (%) | (%) |
| Heptachlor | 13.44 | 0.119 | 0.361 | 0.095 | 0.288 | 73 ± 12 | 18.8 |
| Heptachlor exo-epoxide | 15.67 | 0.140 | 0.424 | 0.148 | 0.449 | 72.5 ± 5.6 | 8.9 |
| Heptachlor endo-epoxide | 15.81 | 0.117 | 0.356 | 0.036 | 0.108 | 79.8 ± 9.9 | 11.7 |
| Malathion | 14.32 | 0.076 | 0.231 | 0.092 | 0.279 | 108 ± 12 | 13.9 |
| Parathion-ethyl | 14.73 | 0.085 | 0.258 | 0.083 | 0.250 | 100.2 ± 9.2 | 12.1 |
| Permethrin | 23.85 | 0.050 | 0.152 | 0.091 | 0.273 | 95 ± 11 | 17.7 |
| Phenothrin | 22.19 | 0.055 | 0.167 | 0.163 | 0.495 | 91 ± 19 | 22.3 |
| Profenofos | 17.38 | 0.129 | 0.391 | 0.098 | 0.296 | 100.9 ± 7.2 | 9.6 |
| Pyrazophos | 23.21 | 0.198 | 0.599 | 0.054 | 0.163 | 99.1 ± 6.6 | 9.0 |
| Pirimiphos-methyl | 14.06 | 0.031 | 0.094 | 0.138 | 0.418 | 88.9 ± 4.4 | 17.6 |
| Quintozene | 11.54 | 0.067 | 0.204 | 0.018 | 0.053 | 87.4 ± 8.1 | 15.4 |
| Tecnazene | 9.13 | 0.071 | 0.214 | 0.081 | 0.246 | 76.5 ± 7.3 | 7.6 |
| Triazophos | 19.30 | 0.135 | 0.410 | 0.126 | 0.381 | 103.8 ± 6.6 | 11.3 |

[a] Retention time; tolerance range ± 0.5%. [b] LOD and LOQ values referred to standard solutions prepared in solvent. [c] LOD and LOQ values evaluated by analyzing spiked samples. [d] Mean values ± standard deviations (n = 6).

Method selectivity was tested by the analysis of 20 independent blank samples. The absence of interfering peaks in the retention time window of interest within a time tolerance of 0.2 min was checked for each analyte by comparing the chromatographic profiles obtained for blank and spiked samples.

The trueness of measurements was assessed in accordance with Decision 2002/657/EC through the analysis of spiked samples, prepared starting from blank material by additions of known amounts of the analytes. Precision and recovery were determined by performing tests on two sets of blank wheat (six replicates each), fortified at a concentration of 0.025 mg kg$^{-1}$ for each analyte. Recovery percentages were calculated by comparing the concentration of spiked samples, calculated through the calibration line, with the nominal fortification level. It was verified that the calculated mean recovery for each pesticide complied with the recovery range of 70–120%, reported in the official documents (European Commission SANTE 2017/11813/EC) dealing with the method validation and quality control procedures for Pesticide Residues Analysis in Food and Feed. The intra-day RSDr values were well below the reference values of 20%, derived by Horwitz equation [29], under repeatability conditions, demonstrating a good method precision.

For the evaluation of uncertainty of analytical results, the metrological approach was adopted, using the validation data obtained from each step of the analytical procedure [30]. Taking into consideration the uncertainties propagation law, the concentration relative uncertainty has been calculated for each pesticide, by the analyte concentration in the spiked sample, the volume of the final extract, and the sample weight before extraction and clean-up. Then, the determination of the measurement uncertainty was performed by considering four sources of uncertainty: (a) preparation of the standard; (b) method reproducibility; (c) method recovery; (d) instrumental calibration curve. A relative expanded measurement uncertainty was calculated using a coverage factor k of 2, corresponding approximately to a 95% confidence level. The recovery data and the uncertainty measurements evaluated for each pesticide are reported in Table 2. As an example, in Figure 1 the chromatographic separation of a multi-analyte standard solution is shown. A good separation was achieved with symmetrical and narrow peaks in the retention time window between 9 and 28 min.

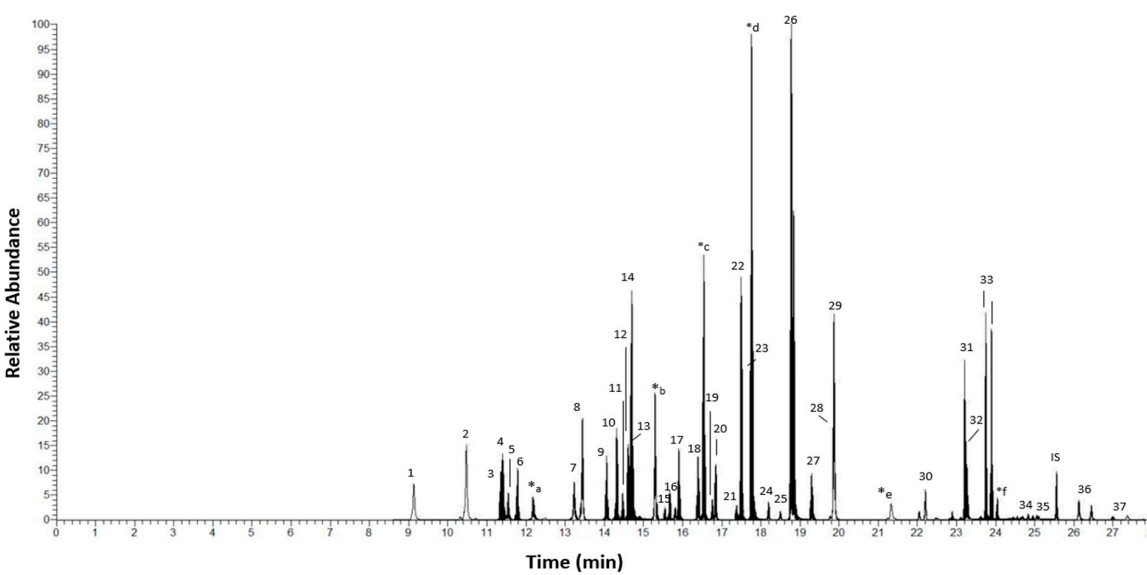

**Figure 1.** Chromatographic separation of a mixed standard solution of (1) tecnazene, (2) α-HCH, (3) β-HCH, (4) γ-HCH, (5) quintozene, (6) diazinon, (7) chlorpyrifos-methyl, (8) heptachlor, (9) pyrimiphos-methyl, (10) malathion, (11) aldrin, (12) fenthion; (13) chlorpyrifos, (14) parathion-ethyl, (15) heptachlor exo-epoxide; (16) heptachlor endo-epoxide, (17) chlorfenvinphos, (18) trans-chlordane, (19) α-endosulfan, (20) cis-chlordane, (21) profenofos, (22) p,p' DDE, (23) dieldrin, (24) endrin; (25) β-endosulfan, (26) p,p' DD, (27) triazophos, (28) endosulfan sulfate, (29) p,p' DDT, (30) phenothrin, (31) pyrazophos, (32) azinphos-ethyl, (33) permethrin, (34) β-cyfluthrin, (35) cypermethrin, (36) fenvalerate, (37) deltamethrin at a concentration of 250 μg L$^{-1}$. IS: internal standard (PCB 209). Star peaks (*a: γ-HCH; *b: pyrimiphos-ethyl; *c: o,p' DDE; *d: o,p' DDD; *e: phosmet; *f: coumaphos) are pesticides not included in the present monitoring study.

### 3.2. Contamination Grade by Pesticide Residues in Cereals and Legumes

The contamination grade by three different classes of pesticides (pyrethroids, organophosphorus and organochlorine compounds) was determined on a total of 220 samples of cereals and legumes, 55% of them coming from Italy. As shown in detail in Figure 2, the other foreign countries belong to Central Asia, Eastern Europe, South Africa, and Canada. Central Asia is one of the major players in international wheat production and Kazakhstan is included in the top ten list of wheat-producing countries and grain exporters in the world [31].

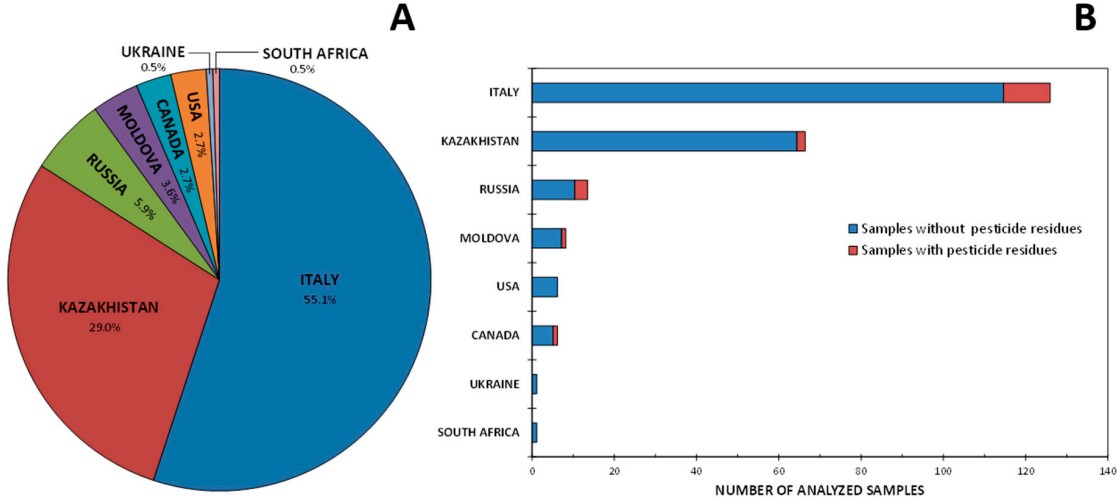

**Figure 2.** (**A**) Origin of cereal and legume samples collected during the years 2018 and 2019 and (**B**) number of contaminated samples.

No trace of pesticides was observed in the analyzed legume samples. Among cereals, the contamination percentage was 7%. Indeed, the presence of one or more pesticides was observed in 18 samples of cereals (16 wheat samples, 1 durum wheat bran sample, and 1 oatmeal sample) at a concentration higher than the corresponding quantification limit, but never exceeding the MRL set by the European Community. Five pesticides, all of them belonging to the class of pyrethroids (cyfluthrin, cypermethrin, deltamethrin, fenvalerate, and phenothrin), were detected in a wheat sample coming from Russia. Nevertheless, this product was compliant with the current EU legislation, since the presence of each of the residues did not exceed the single MRL set for each substance [32].

Analytical results divided by year, type of contaminated sample, and corresponding pesticide level and MRL are displayed in Table 3. PYRs and OPPs were the most present pesticides in cereal samples. The residue of cyfluthrin, deltamethrin, phenothrin, cypermethrin, and fenvalerate (in the class of PYRs) and chlorpyrifos and pirimiphos-methyl (among OPPs) were observed with a concentration content in the range $0.011$–$0.113$ mg kg$^{-1}$, rather below their MRLs, ranging from $0.04$ to $5.00$ mg kg$^{-1}$. The most frequently observed residue among the analyzed cereal samples was pyrimiphos-methyl, which was found not only in wheat samples but also in bran and oats. In 2018, the presence of pyrethroids and organophosphorus compounds was observed in the grain matrix alone, while in 2019 two samples of durum wheat bran and oat flakes were found to be contaminated by pyrimiphos-methyl.

**Table 3.** Contaminated samples by pesticide residues and corresponding MRLs.

| Year | Contaminated Matrix | Pesticide | Contamination Level (mg kg$^{-1}$) | MRL (mg kg$^{-1}$) |
|---|---|---|---|---|
| 2018 | Wheat | Chlorpyrifos | 0.025 | 0.04 |
| | Wheat | Fenvalerate | 0.018 | 0.20 |
| | Wheat | Cyfluthrin | 0.011 | 0.05 |
| | Wheat | Phenothrin | 0.013 | 0.05 |
| | Wheat | Deltamethrin | 0.012 | 1.00 |
| | Wheat | Deltamethrin | 0.018 | 1.00 |
| | Wheat | Cypermethrin | 0.018 | 2.00 |
| | Wheat | Cypermethrin | 0.081 | 2.00 |
| | Wheat | Pirimiphos-methyl | 0.013 | 5.00 |
| | Wheat | Pirimiphos-methyl | 0.030 | 5.00 |
| | Wheat | Pirimiphos-methyl | 0.049 | 5.00 |
| | Wheat | Pirimiphos-methyl | 0.093 | 5.00 |
| | Wheat | Pirimiphos-methyl | 0.113 | 5.00 |
| 2019 | Wheat | Pirimiphos-methyl | 0.015 | 5.00 |
| | Wheat | Pirimiphos-methyl | 0.016 | 5.00 |
| | Wheat | Pirimiphos-methyl | 0.018 | 5.00 |
| | Wheat | Pirimiphos-methyl | 0.045 | 5.00 |
| | Wheat | Pirimiphos-methyl | 0.052 | 5.00 |
| | Wheat | Pirimiphos-methyl | 0.054 | 5.00 |
| | Wheat | Pirimiphos-methyl | 0.030 | 5.00 |
| | Durum wheat bran | Pirimiphos-methyl | 0.080 | 5.00 |
| | Oat flakes | Pirimiphos-methyl | 0.020 | 5.00 |

Figure 2B shows the number of contaminated samples among the analyzed cereals and legumes. A higher number of contaminated samples came from Russia and Canada, with a contamination grade of 23% and 17%, respectively, followed by Moldava with a percentage of contaminated samples equal to 13%. Despite the highest number of analyzed samples coming from Italy and Kazakhstan, their contamination levels resulted quite low, with percentages equal to 9% and 3%, respectively. As an example, the chromatographic profile of a wheat sample contaminated by pyrimiphos-methyl at a concentration of $0.045 \pm 0.008$ mg kg$^{-1}$ is displayed in Figure 3.

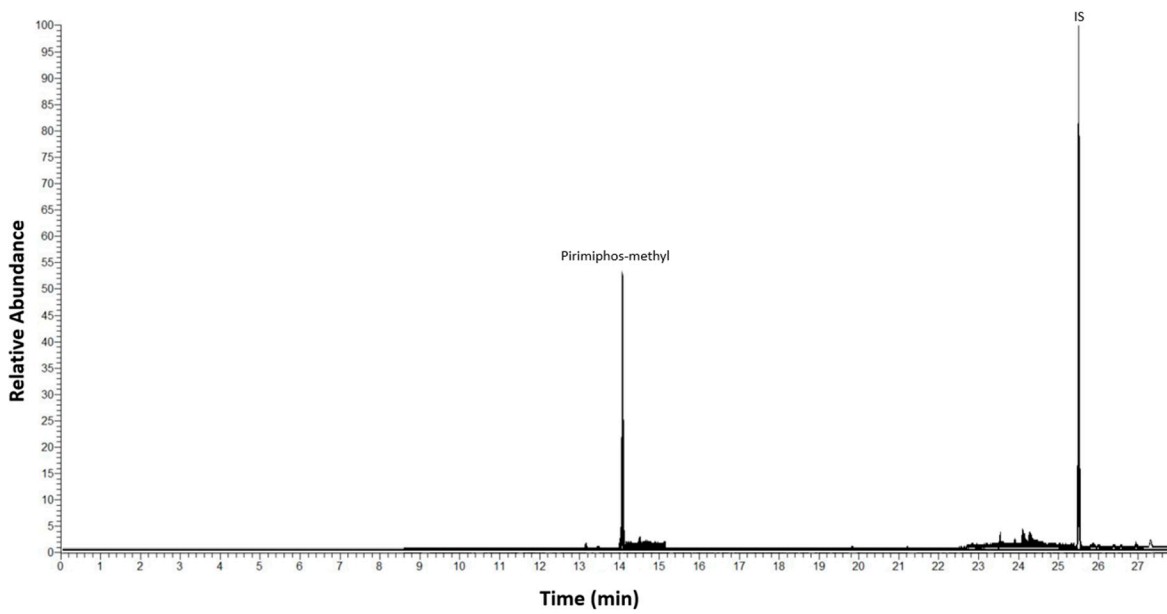

**Figure 3.** Chromatographic separation of a wheat sample contaminated by pyrimiphos methyl.

Our results have been compared with data reported in the official European documents [33], which provides an overview of the contamination grade of cereal samples by pesticide residues in Italy and Europe. According to the national summary reports on pesticide residue analysis performed in 2018, 1457 samples of cereals were analyzed in Italy, of which 90.2% are of Italian origin, 9.8% are imported products. From the analyses carried out on all the cereal samples, it appears that 77.1% of samples did not contain pesticide residues, 22.4% showed residues below or equal to the corresponding MRL, and 0.5% were higher than MRL. Among pesticides listed in the EFSA report whose values exceed MRLs, deltamethrin has been found also in wheat samples monitored in this work, even if at a concentration level lower than its corresponding MRL.

In the European context, 5720 cereal samples were collected in all member states and a percentage of 54.4% showed quantifiable residues. Among them, 48.9% of samples resulted compliantly, and only 5.5% were non-compliant [32]. In the contaminated wheat samples, the following pesticides were identified: chlormequat (39.47%), pirimiphos-methyl (15.08%), mepiquat (8.23), chlorpyrifos-methyl (6.68%), deltamethrin (5.95%) glyphosate (4.75%), cypermethrin (3.81%), tebuconazole (2.58%), dithiocarbamates (1.66%) and chlorpyrifos (1.61%), without exceeding the MRLs. The only residues found with content higher than the corresponding MRLs were chlorpyrifos (0.27%) carbendazim (0.17%) chlorpropham (0.14%) and pirimiphos-methyl (0.13%).

For the extra-European countries, no official data are available on the annual monitoring activity of pesticide residues. Nevertheless, on this topic, an unofficial monitoring study was conducted in Kazakhstan on the determination of pesticide residues in 80 cereal samples (wheat, barley, oats, and rice) [34]. Of the 45 wheat samples, 70% were found to be compliant, associated with non-quantifiable residue levels. A percentage of 16% showed residue contents higher than the quantification limits but lower than the MRL; 13% of samples contained pesticides at concentrations above MRLs. On a total of 15 samples of oats analyzed, 87% were found to be residue-free and 13% with residues below the MRL. A percentage of 86% on 15 barley samples showed no trace of residues and 7% had quantifiable pesticide levels, but lower than MRLs. A further percentage of 7% was associated with samples containing residues above MRLs. The analyses performed of 5 samples of rice showed a percentage of 80% for the residue-free samples and 13% with residues below the MRLs. The detected OPPs are diazinon, malathion, chlorpyrifos-methyl, and pirimiphos-methyl. Among OCPs, residues of DDTs, aldrin, and γ-HCH were found, while the only observed pyrethroid was deltamethrin [34].

### 3.3. Risk Exposure Study

Based on the evidence of the monitoring activity, the evaluation of risk exposure was performed on the following pesticides, in the corresponding raw materials: pirimiphos-methyl in pasta, bread, and breakfast cereals; cypermethrin, cyfluthrin, deltamethrin, fenvalerate, and phenothrin in pasta, chlorpyrifos in bread. Experimental results, obtained in terms of mg die$^{-1}$, NOAEL, and ADI percentages (Tables S1–S3 of Supplementary Data), demonstrated that the estimated risk exposure related to pesticide residues in cereal products such as bread, pasta, and breakfast cereals (evaluated under a high exposure scenario) can be considered very low. Indeed, the percentages of ADI and NOAEL never exceeded values of 0.76% and 0.0086%, respectively. Chlorpyrifos was the pesticide with the highest NOAEL percentage deriving from bread consumption (0.0086% in male adolescents). Pirimiphos-methyl was the most frequently detected pesticide, at the uppermost concentration. Therefore, the highest values of risk exposure deriving from pasta and bread consumption, in terms of ADI percentage, were obtained for this OPP residue. In particular, the highest percentages for bread and pasta consumption were obtained for male adolescents, with values corresponding to 0.76% and 0.64%, respectively. Similar considerations are possible for NOAEL percentages, but referring only to the consumption of pasta (0.0064%). Moreover, pirimiphos-methyl was the only pesticide detected in more than one type of raw material (soft and durum wheat, and oats). Therefore, the risk exposure was also calculated taking into consideration the overall consumption of bread, pasta, and breakfast cereals, within the same day. In this case, the NOAEL and ADI percentages resulted equal to 0.014% and 1.41%, respectively, confirming, also in this case, a very low level of associated risk.

The risk exposure due to some pyrethroid pesticides, such as cyfluthrin, deltamethrin, fenvalerate, and phenothrin, associated with the consumption of pasta, resulted very low, when not negligible. Indeed, NOAEL and ADI percentages of 0.00018% and 0.041% were observed in male adolescents for fenvalerate and deltamethrin, respectively. Slightly higher values were obtained for cypermethrin, with the highest ADI percentage calculated for male adolescents, corresponding to 0.37%, and NOAEL percentages lower than 0.0002%.

## 4. Conclusions

The determination of the contamination grade by pesticide residues in cereals and legumes coming from Italy and foreign countries in the years 2018–2019 was performed through the QuEChERs procedure coupled to GC-MS/MS. The multi-residue pesticide determination allowed to perform high throughput analysis, providing reliable results and quick turnaround of data. The results of method validation demonstrated the conformity of the analytical method with provisions of the European directives for pesticide analysis in monitoring programs, along the food production chain. The evaluation of the contamination grade performed on more than 200 cereal and legume samples highlighted the presence of pesticide residues in the grain samples (with a contamination percentage of 7%), although below the maximum residue levels, while no pesticide was found in the analyzed legumes. PYRs (cyfluthrin, deltamethrin, phenothrin, cypermethrin, and fenvalerate) and OPPs (chlorpyrifos and pirimiphos-methyl) were the most found pesticides in cereal samples (in 18 samples on a total of 209); pirimiphos-methyl was the only residue found not only in wheat samples but also in bran and oats. In terms of risk assessment, pirimiphos-methyl was the pesticide associated with the highest levels of exposure deriving from the consumption of pasta, bread, and breakfast cereals. However, the percentages of exposure are very low, even if evaluated under the highest exposure scenario.

**Supplementary Materials:** The following are available online at https://www.mdpi.com/article/10 .3390/app11167283/s1, Table S1: Risk exposure by pesticide residues of pirimiphos-methyl, cypermethrin, cyfluthrin, deltamethrin, fenvalerate, and phenothrin in pasta under high exposure scenario, Table S2: Risk exposure by pesticide residues of pirimiphos-methyl and chlorpyrifos in bread un-

der high exposure scenario, Table S3: Risk exposure by pesticide residues of pirimiphos-methyl in breakfast cereals under high exposure scenario.

**Author Contributions:** Conceptualization, V.N. and M.Q.; Data curation, M.I. (Marco Iammarino) and D.N.; Formal analysis, M.I. (Marco Iammarino) and F.C.; Funding acquisition, V.N.; Investigation, V.D., M.I. (Mariateresa Ingegno), I.D.R. and F.C.; Methodology, V.D., M.I. (Mariateresa Ingegno), I.D.R. and M.I. (Marco Iammarino); Project administration, V.N.; Resources, V.N.; Supervision, D.L. and M.Q.; Validation, A.C.; Writing—original draft, D.N.; Writing—review & editing, M.Q. All authors have read and agreed to the published version of the manuscript.

**Funding:** This work was supported by Ministero della Salute (Rome, Italy), who financed the pro-ject IZS PB 08/20 RC.

**Institutional Review Board Statement:** Not applicable.

**Informed Consent Statement:** Not applicable.

**Data Availability Statement:** Data available on request.

**Acknowledgments:** Istituto Zooprofilattico Sperimentale della Puglia e della Basilicata (Foggia, Italy) is thanked for providing research support.

**Conflicts of Interest:** The authors declare no conflict of interest.

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
