# Peer review of "Pesticides Contamination of Cereals and Legumes: Monitoring of Samples Marketed in Italy as a Contribution to Risk Assessment"

_applsci, doi:10.3390/app11167283_

Round 1
Reviewer 1 Report
- The methods used for the analysis of these pesticides are not presented, including the standardized methods
- The degree of novelty of the study is not underlined and presented
Author Response
Reviewer n.1:
The authors would like to thank the reviewer for his effort in improving the scientific impact of the Paper. The manuscript has been revised, according to reviewer’s suggestions, editing corrections and rewording the text where necessary.
1) The methods used for the analysis of these pesticides are not presented, including the standardized methods.
Response: Thanks for your comment. A new section has been added at lines 78-85, describing the most used analytical techniques for pesticides determination, together with some references.
2) The degree of novelty of the study is not underlined and presented.
Response: Following the reviewer's suggestion, new sentences regarding the novelty of this study have been added both in the abstract and introduction (lines 42-43 and 97-98).
3) Does the introduction provide sufficient background and include all relevant references? It can be improved.
Reponse: The introduction has been intergrated, adding more comments and underlining the novelty of this study (lines 78-85 and 97-98).
Reviewer 2 Report
The results are solid and presented in a very clean way. The paper reads great. Only one minor comment: I would suggest to include the NOAEL and ADL equations and their specific calculations in the manuscript.Author Response
Reviewer n.2:
The authors would like to thank the reviewer for his effort in improving the scientific impact of the Paper. The manuscript has been revised, according to reviewer’s suggestions, editing corrections and rewording the text where necessary.
1) The results are solid and presented in a very clean way. The paper reads great. Only one minor comment: I would suggest to include the NOAEL and ADL equations and their specific calculations in the manuscript.
Response: Following the reviewer's suggestion, the equation used for calculating NOAEL and ADI percentages has been reported at lines 196-202.
2) English language and style are fine/minor spell check required.
Response: According to the referee's comment, the manuscript has been spell-checked, and some minor errors have been corrected (lines 21, 23, 60, 73, 74, 151, 175, 181, 235, 237, 246, 285, 312, 214 and 345).